# Occurrence of Human Enteric Viruses in Shellfish along the Production and Distribution Chain in Sicily, Italy

**DOI:** 10.3390/foods10061384

**Published:** 2021-06-15

**Authors:** Giusi Macaluso, Annalisa Guercio, Francesca Gucciardi, Santina Di Bella, Giuseppina La Rosa, Elisabetta Suffredini, Walter Randazzo, Giuseppa Purpari

**Affiliations:** 1Istituto Zooprofilattico Sperimentale della Sicilia “A. Mirri”, 90129 Palermo, Italy; giusi.macaluso@izssicilia.it (G.M.); annalisa.guercio@izssicilia.it (A.G.); santina.dibella@izssicilia.it (S.D.B.); Giuseppa.Purpari@izssicilia.it (G.P.); 2Department of Environment and Health, Istituto Superiore di Sanità, 00161 Rome, Italy; giuseppina.larosa@iss.it; 3Department of Food Safety, Nutrition and Veterinary Public Health, Istituto Superiore di Sanità, 00161 Rome, Italy; elisabetta.suffredini@iss.it; 4Department of Preservation and Food Safety Technologies, Institute of Agrochemistry and Food Technology, IATA-CSIC, Av. Agustín Escardino 7, 46980 Paterna, Spain; wrandazzo@iata.csic.es

**Keywords:** enteric viruses, norovirus, hepatitis E virus, hepatitis A virus, PCR, shellfish, food safety, risk assessment

## Abstract

Contamination of bivalve mollusks with human pathogenic viruses represents a recognized food safety risk. Thus, monitoring programs for shellfish quality along the entire food chain could help to finally preserve the health of consumers. The aim of the present study was to provide up-to-date data on the prevalence of enteric virus contamination along the shellfish production and distribution chain in Sicily. To this end, 162 batches of mollusks were collected between 2017 and 2019 from harvesting areas, depuration and dispatch centers (*n* = 63), restaurants (*n* = 6) and retail stores (*n* = 93) distributed all over the island. Samples were processed according to ISO 15216 standard method, and the presence of genogroup GI and GII norovirus (NoV), hepatitis A and E viruses (HAV, HEV), rotavirus and adenovirus was investigated by real-time reverse transcription polymerase chain reaction (real-time-RT PCR), nested (RT)-PCR and molecular genotyping. Our findings show that 5.56% of samples were contaminated with at least one NoV, HAV and/or HEV. Contaminated shellfish were sampled at production sites and retail stores and their origin was traced back to Spain and several municipalities in Italy. In conclusion, our study highlights the need to implement routine monitoring programs along the whole food chain as an effective measure to prevent foodborne transmission of enteric viruses.

## 1. Introduction

Sicily is the Mediterranean Sea’s largest island, and the aquaculture sector plays a significant role in the regional economy. It provides about 4000 tons per year of fishery products, which represents 20% of the overall Italian seafood production [1,2]. The shellfish culture sector is formed by small mussel farms in the Provinces of Palermo, Messina and Syracuse, acting as producers and purification centers for bivalve mollusks to be either locally consumed or exported [3].

The consumption of shellfish products, although recommended for their nutritional values, could pose a risk for human health due to their filter-feeding activity and their capacity to bioaccumulate pathogens from the surrounding waters [4,5]. Among other pathogens, human enteric viruses are excreted in feces by infected people and might reach coastal areas through inefficient wastewater treatments, sewage overflow or direct release of untreated sewage. This factor is particularly concerning in developing countries where untreated sewage is disposed into water sources causing environmental pollution [6]. Contaminated ice or water used in storing or rinsing could determine enteric virus contamination of shellfish along the processing chain as demonstrated by previous epidemiological investigations [7,8].

Hence, shellfish have been proposed as sentinels of the microbial quality of the aquatic environment. In addition, their use for the estimate of the prevalence of viral diseases in population living in coastal areas has been reported for hepatitis A virus (HAV) [9] and more recently hypothesized for SARS-CoV-2 [10]. Norovirus (NoV), rotavirus (RV), adenovirus (AdV), hepatitis A (HAV) and hepatitis E virus (HEV) are the causative agents of gastroenteritis, meningitis and hepatitis. They are all transmitted by the fecal–oral route and represent the viral pathogens of primary concern for waterborne and foodborne outbreaks, including those involving shellfish consumption [11,12,13,14,15,16]. Many reports informed on enteric virus contamination of shellfish products worldwide [17,18]. Moreover, as shellfish are occasionally consumed raw or undercooked, viral particles may survive the most critical inactivation stages during preparation. In Italy, even though the total number of NoV foodborne cases is probably largely underestimated (an average of 72 cases per year is officially notified), the number of foodborne outbreaks showed an increasing trend during the period of the study 2017–2019. Among those, fish and mollusks were the most implicated food commodities, accounting for 45% of the notifications [19,20].

Data on food and water viral contamination in Sicily are scarce, and only one previous survey has comprehensively assessed the prevalence of enteric viruses in shellfish, providing the baseline for the continuous monitoring [21]. Thus, the aim of the present study was to provide up-to-date data on the occurrence of enteric viruses in bivalve species harvested in Sicily, as well as to investigate the viral contamination of shellfish along the entire food chain, including points of sale and restaurants. This approach will deliver a more comprehensive overview on the health risk associated to shellfish consumption in Sicily.

## 2. Materials and Methods

### 2.1. Sampling Sites and Molluscan Shellfish

Sampling of mollusks for viral analyses was performed by local public health inspectors in the framework of the national official monitoring activities primarily aimed at collecting data on compliance to microbiological legislation for shellfish (e.g., *Salmonella* sp. and *Escherichia coli*). A total of 162 fresh and frozen batch samples containing ca. 20–30 shellfish each were collected between April 2017 and September 2019 (2017, *n* = 36; 2018, *n* = 78; 2019, *n* = 48). Specifically, shellfish were collected from three Sicilian harvesting areas classified as B areas according to European legislation (Regulation (EC) No 853/2004), depuration and dispatch centers located in Syracuse (*n* = 42), Palermo (*n* = 6) and Messina (*n* = 15). Mollusks from restaurants (*n* = 6), shellfish markets and retail stores (*n* = 93) were also collected to further investigate the safety along the food chain of Sicilian shellfish (Figure 1).

Mollusks were identified as *Mytilus galloprovincialis* (number of batch samples = 124), *Crassostrea gigas* (*n* = 14), *Ensis directus* (*n* = 1), *Glycymeris glycymeris* (*n* = 2), *Haliotis* sp. (*n* = 1), *Nassarius mutabilis* (*n* = 2), *Ostrea edulis* (*n* = 4), *Ruditapes decussatus* (*n* = 1), *Ruditapes philippinarum* (*n* = 2), *Venus verrucosa* (*n* = 1), *Pahia undulata* (*n* = 1), *Tapes decussatus* (*n* = 1), *Tapes philippinarumm* (*n* = 5), *Donax trunculus* (*n* = 1), *Buccinum undatum* (*n* = 1), *Callista chione* (*n* = 1). All the collected samples were transported under temperature-controlled conditions to the Istituto Zooprofilattico Sperimentale della Sicilia (Palermo, Italy) for analysis.

### 2.2. Enteric Virus Extraction from Molluscan Shellfish

The occurrence of enteric virus in molluscan shellfish was investigated following the approach described in ISO 15216-2 standard method [22]. Briefly, at least 10 mollusks were randomly selected from each sampled batch, their digestive tissues were dissected with a sterile blade, and 2 g of homogenized material spiked with 10 µL of titrated Mengovirus (MgV) clone MC0 as process control (1.6 × 10^4^ TCID_50_/_mL_) (kindly supplied by the Istituto Superiore di Sanità, ISS, Rome, Italy) to monitor extraction efficiency. Then, samples were digested using 2 mL of proteinase K (0.1 mg/mL) at 37 °C for 60 min, and the enzyme deactivated at 60 °C for 15 min. Finally, the supernatants were collected by centrifugation at 3000× *g* for 5 min at 4 °C and retained for nucleic acid extraction. Each series of extractions included a negative extraction control sample that was run through all stages of the analytical process.

### 2.3. Nucleic Acid Extraction and Enteric Virus Detection and Quantification

Viral nucleic acids were extracted from 500 µL supernatants using the NucliSENS miniMAG extraction kit (BioMerieux, Paris, France) according to the manufacturer’s protocol. RNA was eluted in 100 µL and either analyzed immediately or stored at −80 °C until use. Targeted viruses included norovirus GI, norovirus GII, HAV, HEV, AdV and RoV. Norovirus, HAV and HEV were detected by real-time reverse transcription polymerase chain reaction (real-time RT-PCR) using the RNA UltraSense One-Step qRT-PCR System (Invitrogen, Thermo Fisher Scientific, Carlsbad, CA, USA). For RoV analysis, prior to addition of RNA to RT-PCR master mix, sample RNA was subjected to denaturation at 97 °C for 5 min followed by incubation on ice for 2 min, to separate the rotaviral dsRNA. Reverse transcription was performed using Taq DNA Polymerase PCR Buffer, Random Primers and M-MLV Reverse Transcriptase (Invitrogen, Thermo Fisher Scientific, Carlsbad, CA, USA) and PCR was performed using the TaqMan Universal PCR Master Mix (Applied Biosystems, Thermo Fisher Scientific, Carlsbad, CA, USA). Finally, AdV analysis was performed by nested as described below. The set of primers and probes, the targeted regions, the cycling conditions for each molecular assay, along with references are detailed in Table 1.

All the real-time RT-PCR assays were run in a QuantStudio™ 6 Pro Real-Time PCR System instrument (Life Technologies, Thermo Fisher Scientific, Carlsbad, CA, USA). In each real-time RT-PCR run, molecular grade water and a positive target RNA template were included as quality controls. Viral recovery efficiency was calculated for each sample in ratio to the MgV process control virus that was added before extraction. Each sample was also tested for PCR inhibition comparing undiluted and 10-fold diluted RNA quantitation cycle (Cq) threshold for MgV. Confidence intervals (CI_95%_) of the positive results were calculated for proportions.

### 2.4. Viral Genotyping

The genotyping molecular assays were carried out for real-time RT-PCR positive samples using previously published oligonucleotide primers (Table 1). Specifically, a semi-nested RT-PCR for HAV was performed using a GeneAmp RNA PCR Core Kit (Applied Biosystems, Thermo Fisher Scientific, Carlsbad, CA, USA), a broad range nested RT-PCR assay was performed for HEV using MyTaq One-Step RT-PCR Kit and MyTaq Red Mix Kit (Bioline, Meridian Bioscience, Taunton, MA, USA), and a nested PCR for AdV was performed using the Taq PCR Core Kit (QIAGEN, Germany). The positive RT-PCR/PCR products obtained were purified using Illustra GFX PCR DNA and Gel Band Purification Kit (GE Healthcare, North Richland Hills, TX, USA) and were subjected to direct automated sequencing on both strands (BMR Genomics, Padova, Italy). The raw forward and reverse ABI files obtained by sequencing were aligned and assembled into a consensus sequence using MEGA software (version 7, Pennsylvania State University, PA, University Park, USA), and sequences were submitted to BLAST analysis for genotyping at http://blast.ncbi.nlm.nih.gov/Blast.cgi. (accessed on 2 June 2021)

## 3. Results and Discussion

Data about the occurrence of enteric virus along the Sicilian shellfish chain are scarce [21,33], therefore this work expands the knowledge currently available on the safety of food products produced and marketed on the largest Mediterranean island. Specifically, this longitudinal study shows that bivalve molluscan shellfish harvested and marketed in Sicily, Italy, are occasionally contaminated by human pathogenic enteric viruses. We analyzed 162 shellfish batch samples collected between April 2017 and September 2019 and enteric viral contamination was detected by real-time RT-PCR throughout the sampling period. Viral recovery was confirmed by MgV process control that was above 1% in all samples. Similarly, by comparing undiluted and 10-fold diluted MgV RNA Cq thresholds, the occurrence of significant PCR inhibition (∆Cq ≤ 2) was excluded. Thus, our results were validated according to ISO 15216-2 criteria [22]. The adoption of the ISO 15216-2 method in both present (2017–2019) and previous (2012–2017) [21] shellfish surveillance studies conducted on the island allows to study enteric virus contamination over time. Furthermore, it is worth mentioning that the two surveys determined the occurrence of enteric virus by testing a similar size of shellfish samples.

Out of the 162 batch samples, 9 were positive for enteric viruses representing an overall contamination rate of 5.56% (CI_95%_ 2.02–9.08%) (Table 2). All contaminated mollusks belonged to the species *Mytilus galloprovincialis*. This result is not surprising given both the size biased sampling (*M. galloprovincialis* = 124 vs. others = 38, see Table 3) and the specialized production of Sicilian farms that is almost exclusively constituted by mussels [3]. Moreover, the proximity of harvest locations to cities (e.g., Syracuse, Palermo and Messina) that contaminate coastal waters because of their anthropogenic activities could have determined the contamination of those filter-feeding animals able to bioaccumulate viral pathogens.

The detection of enteric virus in shellfish collected in the Mediterranean Sea has been reported with variable contamination rates ranging from zero up to 87% [9,34,35,36,37]. NoVs were the most frequently detected viral contaminant accounting to the 4.32% (CI_95%_ 1.19–7.45%) of all the batches analyzed along the course of the current surveillance study, showing a decreasing trend with respect to 18.5–55% observed in previous works [21,33]. However, it is worth noting that our data may underestimate the real NoV contamination as GIV genotype has been detected previously but not targeted in the current study [21,33]. Six batch samples tested positive for both GI and GII NoV, while an additional sample was positive only for GII NoV. The average GI and GII NoV Ct values were 38.28 ± 1.32 and 36.95 ± 1.50, respectively. Previous reports indicated that NoV prevalence rates in shellfish accounted for 9.8% in Italy [35] and 13% in French coastal regions, Corsica inclusive [34]. According to the most recent EFSA report, NoVs in fishery products are the etiological agents of the highest number of foodborne outbreaks [38]. Only one shellfish batch tested positive for HAV (Ct values 38.63 ± 1.01) representing 0.62% (CI_95%_ 0.58–1.82%) of all the samples. HAV prevalence showed a sharp decrease compared to the 13% (20 out of 108) reported in the previous study [21]. In another study in 2012, 50% of the tested shellfish was positive for HAV, even though such a high prevalence may have been affected by the limited number of samples included in the survey [33]. In Campania, another Region in Southern Italy, a four-years monitoring study reported a 14% HAV prevalence rate being assigned to genotypes 1A (86%) or 1B (14%) [9], in accordance with previous findings in the same area [39]. To avoid an overestimation of the viral occurrence reported, it is worth to note that these surveys were conducted during an HAV outbreak that determined the contamination of harvesting areas. Molecular characterization of the HAV positive sample assigned the sequence to 1B genotype.

An additional batch sample tested positive for HEV with a Cq value of 37.51 ± 0.5. However, the nested RT-PCR assay carried out for HEV genotype assignation yielded negative results. This is not surprising given the poor level of contamination that probably resulted below the sensitivity of the genotyping assay. A prevalence of 1 out of 162 samples, accounting to 0.62% (CI_95%_ 0.58–1.82%), resulted for HEV from the present shellfish surveillance monitoring program, which is similar to the previously reported (1 out of 108, 0.9%). Roughly similar prevalence rates were recently reported in Southern Italy, ranging from 0.89% in Apulia region [40] to 2.6% in Campania Region [9], while previous investigations did not detect HEV contamination in shellfish collected in the Mediterranean basin [33,34,39,41,42]. In contrast, alarming HEV prevalence rates have been reported elsewhere in Europe [43,44,45,46,47], and worldwide [48,49,50], and the consumption of contaminated shellfish has been identified as a risk factor for HEV transmission, although relatively rare [51]. None of the batch samples resulted positive for either RoV or AdV, while contamination was occasionally reported for AdV in a previous study conducted in Sicily (2 out of 108, 1.9%) [21]. In contrast with the year-to-year variation in type of detected enteric virus, we revealed that shellfish contamination was persistent over the period of study, even though at a low prevalence.

In addition, a traceback investigation was conducted to find the source of the shellfish products and to determine where the contamination might have occurred. We were able to trace back all the contaminated samples, being five of them produced in Syracuse, two in Spain, one in Messina and one in Foggia (Italy) (Table 3 and Table 4).

All the three contaminated batches collected at the harvesting sites were locally farmed shellfish belonging to different producers and collected during independent sampling campaigns in Syracuse (May 2017, February and April 2019). Some studies directly linked viral shellfish contamination to sewage spill over into coastal areas [52,53]. Given the proximity of harvest areas to cities, these events may explain the product contamination. The diverse prevalence rates detected among harvesting sites may be due to biased sampling sizes among locations (see Figure 1).

Two additional virus positive batches were collected at depuration centers located in Syracuse (April 2019) and Messina (April 2018). Four more contaminated samples were collected at retail stores located in Palermo (*n* = 2, March and April 2019) and Catania (*n* = 2, February 2018), the two main cities of the island (Table 3 and Table 4). Interestingly, all the four samples were positive for both GI and GII NoV being two batches locally farmed in Syracuse and Messina, and two imported from Spain. The HAV virus-positive sample was produced and collected in Syracuse, while the HEV virus-positive sample originated from the southern part of Apulia region (Foggia) and was collected in a depuration center in Messina (Table 4). Interestingly, the HEV contaminated mussel originated from the same harvesting area and sampling period of an oyster notified as HEV virus-positive by La Bella and colleagues and collected at retail [40].

Even though the presented surveillance study was conducted over a 3-year period, the limited number of virus-positive samples cannot be used to describe a seasonality trend for contaminated shellfish. While it has been reported in other European countries such as France, Ireland and the UK [19,54,55], previous data from Italy are conflicting [36,56].

Finally, none of the six batch samples of raw shellfish collected in restaurants tested positive for any of the targeted viruses. Given the different dates and sites of shellfish production and sampling retrieved from the traceability investigation, the potential cross-contamination that could have been occurred during handling, depuration, sampling or bisection could be excluded.

Overall, this information further confirms the risk for consumers to be exposed to enteric virus infections through shellfish consumption [11].

In summary, the present survey highlights that shellfish commercially available in Sicily presents risk for consumers to be exposed to enteric virus infections, that is of particular concern in the case of HEV. Together with the turnaround time for analytical results on viral contamination, the short shelf life of shellfish represents the main obstacle to timely implement control measures for ensuring consumer health (e.g., product recall). Thus, the most effective measures are preventive strategies that rely on the monitoring of fecal and viral contamination in shellfish harvesting areas and throughout the entire food chain. These actions should be reinforced by both industrial actors and authorities, thus the implementation of a local network able to conduct laboratory analyses and providing assessment results of key importance [19]. On the other hands, consumers and food services should follow the official guidelines provided by the food safety authorities that recommend cooking shellfish for 90 °C for 90 s [20,57] or equivalent time-temperature combinations [58,59,60].

## 4. Conclusions

This study underlines the need to program a monitoring system for enteric viruses through shared molecular methods, associated with epidemiological studies.

Nonetheless, bivalve biomonitoring is effective to guarantee food safety and precisely estimate the risks associated with the consumption of contaminated shellfish.

Following 882/2004 and 854/2004 European Regulations (Annex VII), Italian Authorities included HAV and NoV in national monitoring programs of fruit, vegetables and fish products. In 2018, a signal in this direction is the inclusion of HAV and NoVs into the Integrated Regional Plan of Official Controls of the Sicily Region (PRIC), setting the basis of a coordinated network of local laboratories able to provide rapid and point-of-need response in case of foodborne outbreaks.

## Figures and Tables

**Figure 1 foods-10-01384-f001:**
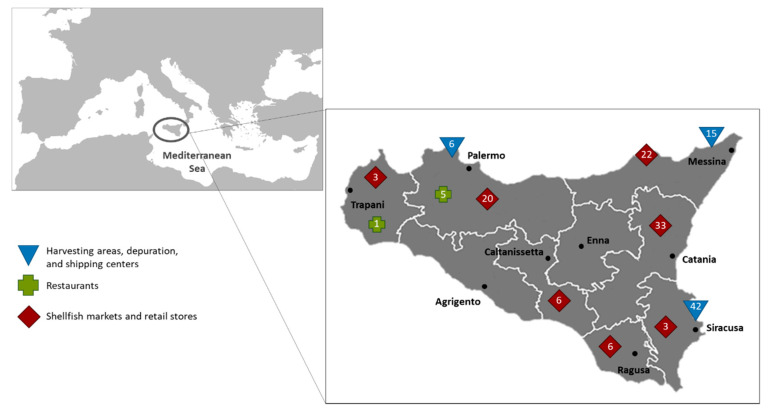
Sampling sites selected to monitor the levels of enteric virus contamination in shellfish along the Sicilian food chain during the period of the study, 2017–2019. Numbers in the symbols indicate the batches of shellfish collected at each sampling location.

**Table 1 foods-10-01384-t001:** Molecular assays used in this study to detect and genotype enteric viruses in shellfish. The sets of primers and probe, targeted regions and cycling conditions are specified for each viral target and molecular assay.

Viral Target	Molecular Method	Primers, Probes	Sequence 5′–3′	Target	Amplification Conditions	PCR Product Length	Reference
HAV	*Semi-Nested* *RT-PCR*	AV1 (Rev)	5′-GGAAATGTCTCAGGTACTTTCTTTG-3′	VP1	RT: 42 °C 60 min, 94 °C 5 min; First: 94 °C 5 min, 40× (94 °C 35 s, 55 °C 1 min, 72 °C 1 min 15 s), 72 °C 5 min. Semi-Nested: 94 °C 5 min, 30× (94 °C 30 s, 50 °C 30 s, 72 °C 30 s), 72 °C 5 min.	First: 247 bp Semi-Nested: 210 bp	[23]
		AV2 (Fw)	5′-GTTTTGCTCCTCTTTATCATGCTATG-3′				
		AV3 (Rev)	5′-TCCTCAATTGTTGTGATAGC-3′				
	*real-time * *RT-PCR*	HAV68 (Fw)	5′-TCACCGCCGTTTGCCTAG-3′	5′-NCR	55 °C 60 min, 95 °C 5 min, 45× (95 °C 15 s, 60 °C 60 s, 65 °C 60 s).	173 bp	[22,24]
		HAV240 (Rev)	5′-GGAGAGCCCTGGAAGAAAG-3′				
		HAV150p (Probe)	FAM 5′-CCTGAACCTGCAGGAATTAA-3′ MGB				
HEV	*real-time * *RT-PCR*	JVHEVF(Fw)	5′-GGTGGTTTCTGGGGTGAC-3′	ORF3	50 °C 60 min, 95 °C 5 min; 45× (95 °C 15 s, 60 °C 60 s, 65 °C 60 s)	69 bp	[25,26,27]
		JVHEVR(Rev)	5′-AGGGGTTGGTTGGATGAA-3′				
		JVHEVPmod (Probe)	5′-FAM-TGATTCTCAGCCCTTCGC-3′-MGB				
	*Nested* *RT-PCR*	ORF1F-1679-f (Fw)	5′-CCAYCAGTTYATHAAGGCTCC-3′	ORF1	RT: 45 °C 20 min; First: 95 °C 1 min, 40× (95 °C 10 s, 51 °C 10 s, 72 °C 30 s), 72 °C 5 min. Nested: 95 °C 1 min, 35× (95 °C 15 s, 48 °C 15 s, 72 °C 10 s), 72 °C 5 min.	First: 348 bp Nested: 172 bp	[28]
		ORF1R-1680-r (Rev)	5′-TACCAVCGCTGRACRTC-3′				
		ORF1FN-1681-f (Fw)	5′-CTCCTGGCRTYACWACTGC-3′				
		ORF1RN-1682-r (Rev)	5′-GGRTGRTTCCAIARVACYTC-3′				
NoV GI	*real-time * *RT-PCR*	QNIF4 (Fw)	5′-CGCTGGATGCGNTTCCAT-3′	ORF2	55 °C 60 min, 95 °C 5 min, 45× (95 °C 15 s, 60 °C 60 s, 65 °C 60 s).	86 bp	[22,29]
		NF1LCR (Rev)	5′-CCTTAGACGCCATCATCATTTAC-3′				
		NVGG1p (Probe)	FAM 5′-TGGACAGGAGAYCGCRATCT-3′ TAMRA				
NoV GII	*real-time * *RT-PCR*	QNIF2 (Fw)	5′-ATGTTCAGRTGGATGAGRTTCTCWGA-3′	ORF2		89 bp	
		COG2R (Rev)	5′-TCGACGCCATCTTCATTCACA-3′				
		QNIFs (Probe)	FAM 5′-AGCACGTGGGAGGGCGATCG-3′ TAMRA				
AdV	*Nested * *PCR*	ADE1–hexAA1885 (Fw)	5′-GCCGCAGTGGTCTTACATGCACATC-3′	hexon genes	First: 94 °C 3 min, 35× (94 °C 30 s, 55 °C 30 s, 72 °C 1 min), 72 °C 5 min. Nested: 94 °C 3 min, 35× (94 °C 30 s, 55 °C 30 s, 72 °C 1 min), 72 °C 5 min.	First: 301 bp Nested: 143 bp	[30]
		ADE2–hexAA1913 (Rev)	5′-CAGCACGCCGCGGATGTCAAAGT-3′				
		ADE3–nehexAA1893 (Fw)	5′-GCCACCGAGACGTACTTCAGCCTG-3′				
		ADE4–nehexAA1905 (Rev)	5′-TTGTACGAGTACGCGGTATCCTCGCGGTC-3′				
RoV	*real-time * *RT-PCR*	NVP3-F Deg (Fw)	5′-ACCATCTWCACRTRACCCTC-3′	NSP3	37 °C 60 min, 95 °C 5 min; 50 °C 2 min, 95 °C 1 min, 40× (94 °C 20 s, 60 °C 1 min).	87 bp	[31]
		NVP3-R1 (Rev)	5′-GGTCACATAACGCCCCTATA-3′				
		NVP3 (Probe)	5′-FAM-ATGAGCACAATGTTAAAAGCTAACACTGTCAA-3′-MGB				
MgV	*real-time * *RT-PCR*	Mengo110 (Fw)	5′-GCGGGTCCTGCCGAAAGT-3′	5′-NCR	55 °C 60 min, 95 °C 5 min; 45× (95 °C 15 s, 60 °C 60 s, 65 °C 60 s)	100 bp	[22,32]
		Mengo209 (Rev)	5′-GAAGTAACATATAGACAGACGCACAC-3′				
		Mengo147 (Probe)	5′-FAM-ATCACATTACTGGCCGAAGC-3′-MGB				

Abbreviations: HAV, hepatitis A virus; HEV, hepatitis E virus; NoV GI, norovirus genogroup I; NoV GII, norovirus genogroup II; AdV, adenovírus; RoV, rotavirus; MgV, mengovirus.

**Table 2 foods-10-01384-t002:** Overview of molluscan shellfish sampled and contamination levels of enteric virus along the Sicilian shellfish chain.

Sampling Site	Sampling Year	No.	HAV	HEV	NoV GI	NoV GII
Harvesting areas	2017	27	1	0	0	0
	2018	10	0	0	0	0
	2019	17	0	0	2	2
	Sum:	54	1	0	2	2
Depuration centers	2017	0	0	0	0	0
	2018	4	0	0	0	0
	2019	0	0	0	0	1
	Sum:	4	0	0	0	1
Shipping centers	2017	2	0	0	0	0
	2018	3	0	0	0	0
	2019	0	0	0	0	0
	Sum:	5	0	0	0	0
Restaurants	2017	4	0	0	0	0
	2018	2	0	0	0	0
	2019	0	0	0	0	0
	Sum:	6	0	0	0	0
Shellfish markets and retail stores	2017	3	0	0	0	0
	2018	59	0	0	2	2
	2019	31	0	0	2	2
	Sum:	93	0	0	4	4
Total		162	1 (0.62) ^a^	1 (0.62) ^a^	6 (3.70) ^a^	7 (4.32) ^a^

Abbreviations: HAV, hepatitis A virus; HEV, hepatitis E virus; NoV GI, norovirus genogroup I; NoV GII, norovirus genogroup II. ^a^, Prevalence of shellfish contamination for each viral target.

**Table 3 foods-10-01384-t003:** Traceability of the viral contamination of shellfish produced and marketed along the food chain in Sicily, Italy, from April 2017 to September 2019. Positive samples are indicated in parenthesis and extended data presented in Table 4.

Species	Common Name	Geographical Origin							
		Italy	France	Greece	Spain	Chile	Tunisia	Netherlands	Vietnam
*Crassostrea gigas*	Pacific oyster	4	9	0	0	0	0	1	0
*Mytilus galloprovincialis*	Mediterranean mussel	105 (7)	0	1	16 (2)	2	0	0	0
*Ostrea edulis*	Mud oyster	3	1	0	0	0	0	0	0
*Ruditapes philippinarum*	Venus clam	2	0	0	0	0	1	0	0
*Tapes decussatus*	Clam	1	0	0	0	0	0	0	0
*Tapes philippinarumm*	Venus clam	4	1	0	0	0	0	0	0
*Buccinum undatum*	Common whelk	0	1	0	0	0	0	0	0
*Donax trunculus*	Wedge clam	0	1	0	0	0	0	0	0
*Glycymeris glycymeris*	Dog cockle	0	2	0	0	0	0	0	0
*Haliotis* sp.	Abalone	0	1	0	0	0	0	0	0
*Nassarius mutabilis*	Mutable nassa	0	2	0	0	0	0	0	0
*Ensis directus*	Jackknife clam	0	0	0	0	0	0	1	0
*Venus verrucosa*	Warty venus	0	0	0	1	0	0	0	0
*Paphia undulata*	Pahia undulata	0	0	0	0	0	0	0	1
*Callista chione*	Smooth clam	1	0	0	0	0	0	0	0
Total		120	18	1	17	2	1	2	1

**Table 4 foods-10-01384-t004:** Detailed traceability data of the viral contaminated shellfish resulted from the study. All virus-positive samples were identified as *Mytilus galloprovincialis*.

Sample ID	Sampling Date	Sampling Site Type	Sampling Site Location	Geographical Origin	Viral Contamination
1	22 May 2017	Harvesting area	Syracuse (Italy)—Site A	Syracuse (Italy)	HAV 1B
2	5 February 2018	Shellfish market/retail store	Catania (Italy)	Spain	GI and GII NoV
3	6 February 2018	Shellfish market/retail store	Catania (Italy)	Spain	HEV
4	1 April 2018	Depuration center	Messina (Italy)	Foggia (Italy)	GI and GII NoV
5	12 February 2019	Harvesting area	Syracuse (Italy)—Site B	Syracuse (Italy)	GI and GII NoV
6	19 March 2019	Shellfish market/retail store	Palermo (Italy)	Syracuse (Italy)	GI and GII NoV
7	1 April 2019	Harvesting area	Syracuse (Italy)—Site C	Syracuse (Italy)	GI and GII NoV
8	4 April 2019	Shellfish market/retail store	Palermo (Italy)	Messina (Italy)	GI and GII NoV
9	8 April 2019	Depuration center	Syracuse (Italy)	Syracuse (Italy)	GII NoV

Abbreviations: HAV, hepatitis A virus; HEV, hepatitis E virus; NoV GI, norovirus genogroup I; NoV GII, norovirus genogroup II.

## Data Availability

The raw data supporting the conclusions of this article will be made available by the authors, without undue reservation.

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
