# Peer review of "Occurrence of Human Enteric Viruses in Shellfish along the Production and Distribution Chain in Sicily, Italy"

_foods, 2021, doi:10.3390/foods10061384_

Round 1

Reviewer 1 Report

Introduction:

please provide data on the number of human cases of infection with the virus in Italy per annum.

Line 82: “classified” should be “identified”

Line 83: Haliotis sp.

Line 92: it is mentioned 10 molluscs from each batch but then lines 82 to 86, have numbers like 1. Please make sure across the manuscript you specify if you are talking about the number of individual molluscs or individual batch or else.

Author Response

Dear Reviewer,

We acknowledge your efforts in providing useful comments in such a timely manner. Please find below a point-by-point response covering all the observations raised by the reviewers. 

Reviewer 2 Report

This paper reports a study human enteric viruses in shellfish along the production and distribution chain in Sicily. The paper provides some useful information. Minor modification will be required to the following points.

  1. Figure 1: The longitude and latitude of the sample collection place
  2. Table 1 : Abbreviations need to be marked
  3. Table 3 : The common name of the test sample must be marked.
  4. Conclusions: Please add that since the seafood products in Italy have an infection rate of 5.56%, why is it higher than that of other countries (Table 3)? Suggest how local Italian consumers should handle or cook such infected food?

Author Response

(The authors gave the same response as above.)

Reviewer 3 Report

Please find the review attached.

Author Response

(The authors gave the same response as above.)

Round 2

Reviewer 3 Report

The new text sections improved the manuscript and various other edits provided more clarity. There are a few text errors in those new sections that I think are worthy of correcting: 

Line 65: "Among those, fish and molluscs were the food vehicles most implicated accounting to the 45% of the notifications (EFSA, and ECDC. 2021; EFSA, Biological hazards reports database)." 

Better something like this: Among those, fish and molluscs were the most implicated food commodities, accounting for 45% of the notifications (EFSA, and ECDC. 2021; EFSA, Biological hazards reports database).

Also, could you please check the references? Is it: EFSA and ECDC 2021; ?

Line 151: April

Line 182: size-biased // vs. (non-italic and a period)

Line 185: e.g., Syracuse, Palermo, and Messina) 

Line 255: "may be due to the biased size sampling among": Better: due to biased sampling sizes among

Line 256: virus-positive

Line 271: in other European countries

Line 289: The short shelf life of shellfish, together.... viral contamination, represent...

Or even better: Together with the turnaround time for analytical results on viral contamination, the short shelf life of shellfish represents the main obstacle to timely implement control measures for ensuring consumer health (e.g., products recall).

Line 293: (e.g., product recall)

Line 300: recommend cooking shellfish

Author Response

Dear Editor,
We acknowledge again reviewers’ efforts in providing additional comments to fine-tuning the manuscript. Please find below a point-by-point response covering all the observations raised by Reviewer 3.
